# Simultaneous Improvement in the Thermostability and Catalytic Activity of Epoxidase Lsd18 for the Synthesis of Lasalocid A

**DOI:** 10.3390/ijms242316795

**Published:** 2023-11-27

**Authors:** Ning Liu, Hongli Xiao, Yongjian Zang, Longji Zhou, Jun Mencius, Zhiwei Yang, Shu Quan, Xi Chen

**Affiliations:** 1Key Laboratory of Synthetic and Natural Functional Molecule of the Ministry of Education, College of Chemistry and Materials Science, Northwest University, Xi’an 710127, China; liuninglnnl@163.com (N.L.); xiaohongli@stumail.nwu.edu.cn (H.X.); 2MOE Key Laboratory for Nonequilibrium Synthesis and Modulation of Condensed Matter, School of Physics, Xi’an Jiaotong University, Xi’an 710049, China; zyj198984@stu.xjtu.edu.cn (Y.Z.); yzws-123@xjtu.edu.cn (Z.Y.); 3Institute of Physics and Electronic Information, Yunnan Normal University, Kunming 650504, China; 4State Key Laboratory of Bioreactor Engineering, East China University of Science and Technology, Shanghai 200237, China; y85220061@mail.ecust.edu.cn (L.Z.); acemencius@gmail.com (J.M.)

**Keywords:** natural product, Lsd18, thermostability, FoldX, Rossetta-ddG, catalytic activity

## Abstract

Enzymes used in the synthesis of natural products are potent catalysts, capable of efficient and stereoselective chemical transformations. Lsd18 catalyzes two sequential epoxidations during the biosynthesis of lasalocid A, a polyether polyketide natural product. We performed protein engineering on Lsd18 to improve its thermostability and catalytic activity. Utilizing structure-guided methods of FoldX and Rosetta-ddG, we designed 15 mutants of Lsd18. Screening of these mutants using thermal shift assay identified stabilized variants Lsd18-T189M, Lsd18-S195M, and the double mutant Lsd18-T189M-S195M. Trypsin digestion, molecular dynamic simulation, circular dichroism (CD) spectroscopy, and X-ray crystallography provided insights into the molecular basis for the improved enzyme properties. Notably, enhanced hydrophobic interaction within the enzyme core and interaction of the protein with the FAD cofactor appear to be responsible for its better thermostability.

## 1. Introduction

Natural products are primary and secondary metabolites produced by living organisms [1], and they are widely utilized as therapeutics, especially for cancer and infectious diseases [2]. Enzymes that synthesize natural products are nature’s chemists, which possess the advantages of simplicity, selectivity, safety, and sustainability [3]. Hence, their application in organic and industrial synthesis is highly desirable. However, wild-type enzymes are often unsuitable for commercial application due to their lack of stability at a low or high pH, elevated temperature, or high organic cosolvent [4]. To overcome such issues, protein engineering approaches are implemented to achieve improved enzyme stability.

Lasalocid A, a polyether ionophore natural product derived from *Streptomyces lasalocidi*, is a widely used veterinary medicine that acts by transporting various cations across the cell membrane, thereby disrupting cation homeostasis [5]. The total synthesis and the biosynthesis pathway of Lasalocid A have been reported [6,7]. Lasalocid A biosynthesis entails two stereospecific epoxidation of the precursor diene, catalyzed by the flavin-dependent enzyme Lsd18 [8]. Epoxides, known for their reactivity, hold great promise as intermediates when constructing pharmaceutical compounds [9]. In organic synthesis, the enantioselective epoxidation of alkenes remains a long-standing objective. The application of biological enzymes, such as Lsd18, provides an opportunity for the efficient synthesis of epoxides.

As a wild-type natural enzyme, Lsd18 has not been engineered before. As a result, improving its thermostability is the first step to promoting its application in the organic and industrial synthesis of epoxides. In this study, combining structural biology and bioinformatics, we applied the tools of FoldX and Rossetta-ddG to design mutants of Lsd18 in order to increase its thermostability. We cloned 15 Lsd18 mutants, 9 of which were successfully expressed and purified. Two mutants, T189M and S195M, along with their double mutants, showed significantly increased thermostability. The underlying mechanism of this improvement was revealed by molecular dynamics simulations and crystallography. Our study represents the first instance of engineering an epoxidase for enhanced thermostability, introducing new routes for achieving this goal.

## 2. Results

### 2.1. Rational and Semi-Rational Designs

To rationally design the mutants of Lsd18 with increased thermostability, we first employed the FoldX algorithm to assess the effect of mutations on the overall stability of Lsd18 based on the X-ray crustal structure of Lsd18 (PDB: 8UP4). The change in the Gibbs free energy of folding (ΔΔG) for every possible amino acid substitution at each position was calculated. Mutations were then selected based on the rank order of the calculated ΔΔG values, along with the following criteria: (a) substitutions of residues involved in FAD binding or catalysis were excluded; (b) proline appearing within secondary structure elements were excluded; (c) if multiple substitutions at the same position were predicted to have similar ΔΔG values, then a visual inspection of the local structures was used to guide mutation selection. Overall, we selected nine mutations, with the calculated ΔΔG values ranging from −4.59 to −2.57, for further analysis (Table 1).

To enhance the precision of mutation stability predictions, a common practice involves the combined utilization of Rosetta-ddG and FoldX algorithms [10]. Therefore, we also utilized Rosetta-ddG to compute ΔΔG values for all saturation mutagenesis variations of the amino acids within Lsd18. As shown in Table 2, Rosetta-ddG identified three additional mutants with relatively large ΔΔG values, making them as potential candidates for further analysis.

In addition to the aforementioned rational designs, we also incorporated semi-rational designs into our approach. Based on the crystal structure of Lsd18 in complex with substrate 1 (PDB: 8WVT), we identified residues I72, V252, V342, and Y218, which locate in the active site and directly interact with the substrate, thus playing a pivotal role in catalysis. To maintain the integrity of these important residues, we focused our attention on residues H239, S237, and S108, which are in close proximity to the substrate but exhibit relatively weak interactions. These hydrophilic residues were intended to be mutated into hydrophobic residues to enhance their interactions with the hydrophobic substrate compound 1. Following the calculation of ΔΔG by FoldX (Appendix A), it was determined that mutants S108V, S237V, and H239L would not lead to the destabilization of Lsd18. Therefore, we proceeded to design a triple mutant, denoted as Lsd18-S108V-S237V-H239L, for further investigation.

Another plausible approach to enhance Lsd18′s stability is to stabilize the FAD bound in Lsd18 because FAD serves as a vital cofactor for Lsd18. Consequently, we propose that mutations to the residues in Lsd18 that are in close proximity to FAD might potentially increase the stability of Lsd18. We rationally designed the mutant R51W to introduce π-π stacking interactions between Lsd18 and FAD. Furthermore, we designed mutants D52S and V64D to establish hydrogen bonds between FAD and the protein. Again, based on the ΔΔG values calculated by FoldX (Appendix A), these three mutants would not destabilize Lsd18. Therefore, the triple-mutant Lsd18-R51W-D52S-V64D was also designed for further testing.

### 2.2. Cloning, Expression and Purification of the Designed Mutants

Mutagenesis was performed through overlap PCR. Out of the 15 designed mutants of Lsd18, 12 were successfully expressed (Figure 1A). Notably, we observed a substantial increase in the expression levels of mutants Lsd18-T189M, Lsd18-S195M, and Lsd18-T189M-S195M compared to the wild-type protein. The yield of the three Lsd18 variants from 1 L of *E. coli* was 14.25 mg, 17.8 mg, and 18.6 mg, respectively, which represented 2.1-fold, 2.6-fold, and 2.7-fold increases over the yield of the wild-type protein (Figure 1B). Given that the abundance of soluble proteins often correlates with their thermostability [11,12], we speculate that Lsd18-T189M, Lsd18-S195M, and Lsd18-T189M-S195M are promising candidates for stabilized mutants.

### 2.3. Screening for the Lsd18 Mutants with Increased Thermostability and Catalytic Activity

We applied a thermoshift assay to evaluate the melting temperature (Tm) of the selected Lsd18 mutants. Tm is a well-established indicator of protein thermostability [13]. We found that the Tm values of Lsd18-T189M, Lsd18-S195M, and Lsd18-T189M-S195M were increased by 3.0 °C, 4.5 °C, and 4.8 °C, respectively (Figure 2A).

As another measure of protein stability, we performed limited proteolysis by incubating these Lsd18 variants and the wild-type protein with a low concentration of trypsin (2 μg/mL). As shown in Appendix A, after incubating these proteins with trypsin for 45 min, the wild-type Lsd18 completely disappeared, while some intact proteins remained for the three mutants, especially for Lsd18-S195M and Lsd18-T189M-S195M. Since the three Lsd18 mutants were more resistant to both heat denaturation and protease digestion, we concluded that they have enhanced stabilities compared with the wild-type Lsd18.

It has been reported that the cofactor FAD can enhance the stability of its associated proteins, leading to increased protein levels [14]. We also observed that the yellower the color of Lsd18 (attributed to bound FAD), the more stable it was. Therefore, we tested the content of FAD quantitatively in the three mutants and wild-type Lsd18. As expected, the FAD content in the mutants Lsd18-T189M, Lsd18-S195M, Lsd18-T189M-S195M increased by 29.3%, 46.7%, and 38.3%, respectively, compared to the wild-type Lsd18 (Figure 2B).

Since there is often a trade-off between increased stability of an enzyme and its catalytic activity, we further determined the enzymatic activity of the three mutants. We synthesized compound 1, a natural substrate analog for Lsd18 (Figure 2C,D), and used this substrate to evaluate the activity of these mutants (Figure 2E). At the standard catalytic temperature of 30 °C, the activities of the three mutants were comparable to that of the wild-type Lsd18. However, when the reaction temperature was raised to 33 °C, both Lsd18-T189M and Lsd18-S195M exhibited enhanced activity, showing an approximately 10–15% increase compared to the wild-type enzyme. This suggests that these two mutants can effectively substitute the wild-type enzyme and perform efficiently 33 °C. Furthermore, as the reaction temperature further increased, all three Lsd18 mutants began to display increased activity compared to the wild-type enzyme. At a reaction temperature of 40 °C, Lsd18-S195M exhibited comparable activity to that of the wild-type enzyme at 40 °C. Collectively, these results demonstrate that our engineering strategy enhanced Lsd18′s thermal stability.

### 2.4. Molecular Dynamics Analysis of Lsd18-T189M, Lsd18-S195M and Lsd18-T189M-S195M

In order to verify the thermal stability of the mutant systems, we carried out molecular dynamics simulations of the wild-type Lsd18 system and three mutant architectures (T189M, S195M, and T189M-S195M), as well as a comparative analysis of structural stability. The 200 ns simulation results revealed that the structures of the four systems are relatively stable and converge after 100 ns (Figure 3A). However, the wild-type structure fluctuates greatly compared with the three mutations, T189M and T189M-S195M are relatively stable during the simulations, and S195M is the most stable during the simulations.

Principal component analysis (PCA) was performed to compare the conformational stability of the four systems. For the wild-type system, PC1 captured over 47% of the structural variance, and the second principal component (PC2) accounted for over 15%. The eigenvalues of the first two PCs (PC1 and PC2) captured more than 60% of the motion of the Lsd18 protein. Hence, the first two PCs are sufficient to analyze the dynamic structure of the Lsd18 protein. A subspace made of wild-type PC1 and PC2 was created, and the sampling structures of the three mutation systems were projected on it, as shown in Figure 3B–E, where the color changes from blue to red to reflect the structure’s stability or instability. The wild-type system’s conformation is not stable, and it tends to be in a narrow, low-energy state (−5, −18) for the stable state. The distribution of the sample structure in conformational space demonstrates that the mutation alters Lsd18’s sampling trend. The T189M mutation emerges as one stable region and two metastable regions in the conformational space, and the stable structure of Lsd18 is more widely dispersed in the space, improving the stability of the system. The stability of the system is obviously improved by the S195M mutation, and the captured structure is almost stable in one conformation. The structural stability of the double-mutation system is lower than the S195M mutation system but higher than the T189M mutation system. These results revealed that the mutation could alter the stability of the Lsd18 structure, as well as its sampling trend. The T189M mutation can partially stabilize the Lsd18 protein, while the S195M mutation can improve the structural stability of Lsd18 protein, in contrast to the T189M and double mutations.

### 2.5. Crystallography Study of Lsd18-S195M, Lsd18-T189M-S195M

Before the crystallography study, we also performed CD analyses to compare the secondary structures of Lsd18, Lsd18-T189M, Lsd18-S195M and Lsd18-T189M-S195M. As shown in Figure 4A,B, no apparent differences in the negative peak were observed, which indicated that the mutants were as well-structured as the wild-type protein.

In order to reveal the structural basis for the higher thermostability of the best two mutants, we solved the crystal structures of Lsd18-S195M and Lsd18-T189M-S195M at the resolution of 2.50 Å and 3.76 Å, respectively. Although the resolution of the second structure was not high, the secondary structure was clear enough for analysis. Comparing the structures of Lsd18 WT, Lsd18-S195M, and Lsd18-T189M-S195M, it can be seen that the angle of the α helix between L408 and T419 shifts, which is in agreement with the MD study. However, in the crystal structure of Lsd18-S195M, a loop in this region turns to a helix, indicating the effect of stabilization. Furthermore, the loop region R314-L316, E208-F210 and C398-Q401 in Lsd18 WT change to a β sheet in the two mutants’ crystal structures, also indicating that this region is less flexible and more stable.

The interactions between M189 and M195 and their surrounding residues are the same as in the MD analysis. In wild-type Lsd18, there is a hydrogen bonding network between S195 and D187, S193 and L199. Moreover, weak hydrogen bonds formed between S195 and L198, as well as Y319. After mutating from S195 to M195, the hydrogen bonds involving D187 and Y319 were removed and new hydrophobic interactions formed between M195 and L199, as well as V331. T189 formed hydrogen bonds with D187 and G192. After mutating to M189, the hydrogen bond with G192 was weakened, while there were hydrophobic interactions between M189 and L26 as well as P194. More importantly, M189 forms a hydrogen bond with the cofactor FAD that could stabilize the binding of this cofactor. Therefore, it is speculated that this could improve the enzymatic activity of this mutant.

## 3. Discussion

In this study, we utilized FoldX and Rosetta-ddG to predict mutations that increase the thermostability of Lsd18. The prerequisite for this analysis is our high-resolution structure of Lsd18 (1.5 Å). The result is that two mutants out of the 12 selected potential mutants showed relatively significant increased thermostability. The successful ratio was 1/6, which is not very high. However, the underlying mechanisms of the stabilization provide some hints of this kind of protein engineering. First, the hydrophobic interactions inside the core of the protein structure might be more important than hydrogen bonds for maintaining the stability of the protein. Second, for proteins with FAD as the cofactor, strengthening the interactions between FAD and the protein might further stabilize the whole protein.

Six methods to assess computational methods for predicting protein stability upon mutation have been compared and evaluated [15]. It is noted that mutations located in β sheets were more accurately predicted by methods like CC/PBSA, EGAD, I-Mutant2.0, and Hunter. On the contrary, FoldX and Rosetta demonstrated superior predictive capabilities for mutations within unstructured regions. Indeed, our most effective mutations, T189M and S195M, reside within the loop region of Lsd18, in agreement with the above discussion.

The crystal structures of the mutants showed that stabilization could lead to more rigid enzymes, which might not be beneficial for the activity of the enzyme. However, at a higher temperature, the mutants could overcome this rigidity and be as catalytically active as the wild-type enzyme at a lower temperature. Although there is a dilemma between stability and activity, our example shows that improving the stability of the enzyme could help mutated enzymes to retain their catalytic activity at a higher temperature. Currently, at 33 °C and after 10 h, Lsd18-T189M could convert 51.3% of the substrate to the product, highlighting its potential for catalysis by Lsd18. Future efforts aim to optimize this enzyme further, focusing on enhancing its catalytic efficiency and broadening its substrate range. This optimization could lead to improved enzymatic activity and versatility, opening doors for diverse applications in biocatalysis.

## 4. Materials and Methods

### 4.1. Rational Design of the Mutants

In preparation for FoldX calculations, the PDB file of Lsd18 required a critical adjustment to reverse the mutation involving the ethylated unnatural amino acid at the 300th position. Subsequently, this modified structure underwent a meticulous minimization process using the RepairPDB function within the FoldX5.0. The essential cfg configuration file was generated through the ‘prepare4scan.py’ script, available at ‘https://github.com/JinyuanSun/DDGScan/tree/master/GUI.’, accessed on 2 November 2021. This cfg file contained vital information related to the strategic design of saturation mutations for all residue sites. Finally, we conducted a saturation mutation scan for Lsd18 to obtain ddG (ΔΔG) data using PositionScan of FoldX 5.0. All the options were set to default settings.

To prepare the input structure for Rosetta prediction, all small molecules in the crystal structure of Lsd18 were removed in PyMOL, and any unreasonable torsions and ligands were eliminated by utilizing Rosetta’s built-in Python script ‘clean_read’. Subsequently, effective structural relaxation and energy minimization were performed using the ‘relax.linuxgccrelease’ script, which is compiled with GCC, as provided by the Rosetta package. The cleaned and relaxed structure serves as the input PDB structure for the Rosetta ddg monomer mutational scan. To generate all possible mutations, a Python script was employed to parse the processed amino acids from the PDB file, creating a ‘mutations.resfile’ that corresponds to all twenty amino acid mutations. To expedite the entire mutational scan, we implemented a parallel computation using ‘mpirun -n 6’, to simultaneously predict the effects of mutations on six physical CPU cores. The Rosetta monomer ddg module is configured with ‘ddg_flags’, specifying 50 simulation cycles, while leaving other parameters at their default settings.

### 4.2. Site-Directed Mutagenesis of Lsd18

Overlap PCR was performed to obtain all the mutants. For example, to obtain S195M, first, the full length of the gene was used as a template for PCR, with primers of Lsd18-F-NdeI and S195M-R containing the mutant gene, and primers of Lsd18-R-EcoRI and S195M-F containing the mutant gene. After the first round of PCR, the F-end and R-end were obtained and further used as the templates for the second round of PCR, which use Lsd18-R-EcoRI and Lsd18-R-EcoRI as primers to obtain the full length of the gene with the mutation of S195M. All the genes were inserted into the plasmid of pCold (Takara Bio, Beijing, China) by the restriction sites of NdeI and EcoRI.

Lsd18 mutagenesis primer design can be found in Table 3.

### 4.3. Expression and Purification of Wild-Type and Mutant Enzymes

The plasmid of lsd18 WT and mutants were co-transformed with the chaperone plasmid pG-KJE8 (Takara Bio, Beijing, China) into BL21(DE3). The culture was grown in Luria Broth medium to OD_600_, reaching 0.6. The production of the protein was induced by 100 µM IPTG, 0.5 mg/mL L-arabinose and 5 ng/mL tetracycline. The culture was incubated for another 20 h at 15 °C. Cells were harvested by centrifugation at 6000 rpm for 20 min at room temperature, resuspended in a lysis buffer containing 50 mM NaH_2_PO_4_, 500 mM NaCl, pH = 8.0, and lysed by sonication. After centrifugation at 13,000 rpm for 45 min, the cleared supernatant was mixed with Ni beads (Sangon Biotech, Shanghai, China). The mixture was incubated for 30 min at 4 °C with moderate shaking and was loaded onto a column. The beads were washed with a lysis buffer until there was no protein to wash off and the target protein was eluted with an elution buffer containing 50 mM NaH_2_PO_4_, 500 mM NaCl, 500 mM imidazole, pH = 8.0. The proteins were further purified by gel filtration on a Superdex200 10/300 GL column (GE Healthcare Life Sciences, Umeå, Sweden) in 20 mM Tris, 150 mM NaCl, pH = 8.5. The purity of all the proteins was analyzed by SDS-PAGE.

### 4.4. Thermal Shift Assay

The mixture was prepared according to Table 4. Real-Time PCR (Thermo Fisher Scientific, Singapore) was applied to record the fluorescent curve with the program of 15 ℃ for 5 min and then 85 ℃ for 30 s with an increment of 1%.

### 4.5. Trypsin Digestion Assay

The WT and mutants of Lsd18 (0.4 mg/mL) and trypsin (2 µg/mL, Sangon Biotech, Shanghai, China) were incubated with the buffer containing 20 mM Tris, pH = 8.5 and 10 mM CaCl_2_. After 45 min, 1mM PMSF was added to stop the reaction. The samples were loaded to SDS-PAGE for further analysis.

### 4.6. FAD Content Determination

FAD (Sangon Biotech) was dissolved in 20 mM Tris, pH = 8.5, 150 mM NaCl. 5 µM, 10 µM, 50 µM, 80 µM and 100 µM FAD were prepared and the absorbance at 451 nm was measured to make the standard curve (Appendix A). To measure the FAD content in the Lsd18 protein, the protein sample was heated at 100 °C for 10 min and centrifuged at 15,000× *g*. The supernatant was collected, and the absorbance was measured at the wavelength at 451 nm.

### 4.7. The Enzymatic Assay

The buffer system for this assay is 50 mM Tris, pH = 8.0, 300 mM NaCl, 10% (*v*/*v*) glycerol. All the reagents were added according to Table 5. The enzymatic assays with WT and mutants of Lsd18 were performed at 30 °C, 33 °C, 37 °C and 40 °C with shaking at 220 rpm for 10 h. Then, 200 µL ethyl acetate was added to quench the reaction and the product was extracted to the organic layer. After evaporation, methanol was added to dissolve the product. The synthetic method used to synthesize this substrate was described in another manuscript.

### 4.8. LC-MS Analysis

LC-MS (Agilent 6460 Triple Quad LC/MS, Santa Clara, CA, USA, resolution 0.01 *m*/*z*) was applied with an Agilent ZORBAX Eclipse XDB-C18 column at a flow rate of 0.2 mL/min and column temperature of 30 °C. Buffer A was 99.9% H_2_O and 0.1% formic acid, and buffer B was 99.9% methanol and 0.1% formic acid. Mass spectra were collected in positive ion mode 100–1000 *m*/*z*. The program was set as shown in Table 6. For the epoxidation product, peak 508.3+ *m*/*z* was selected for analysis. The results are shown in Appendix A.

### 4.9. CD Analysis

The purified Lsd18 WT and mutant proteins were diluted to 1 μM and incubated on ice. The sample was then transferred to the 1 cm glass cuvette and UV (200–500 nm) was measured by CD spectrometry (J-1500) (Easton, MD, USA).

### 4.10. Molecular Dynamics Simulations

The initial coordinates were derived from the structure of wild-type Lsd18 protein, The Lsd18-T189M, Lsd18-S195M and Lsd18-T189M-S195M systems were prepared by mutating T189M, S195M and T189M-S195M, respectively. All-atom explicit-solvent MD simulations were performed using the AMBER18 software package [16]. The ff14SB force field [17] and generalized AMBER force field (GAFF) [18] were chosen to describe the atomic interactions of Lsd18 and FAD. All systems were solvated and placed in a cubic box, with a minimal distance of 10 Å between any system atom and the box wall. Three Na^+^ were added to neutralize the systems. To remove the bad contact between initial structures, each system was initially minimized. After minimization, the systems were heated gradually from 0 to 310 K within 1.0 ns, with a positional restraint of 20.0 kcal/mol∙Å^2^ in protein. This was followed by constant-temperature equilibration at 310 K for 1.0 ns, with a positional restraint of 10.0 kcal/mol∙Å^2^ on protein atoms in a canonical ensemble (NVT). Subsequently, an equilibration simulation was performed for 1.0 ns at 1 atm and 310 K in the NPT ensemble with a positional restraint of 4.0 kcal/mol∙Å^2^ on backbone atoms of the protein [19]. The production MD simulations were carried out at 310 K and 1 atm with a time step of 2.0 fs. Finally, the 200 ns production simulation was performed on each system.

### 4.11. X-ray Crystal Structure Determination

To facilitate crystallization, reductive ethylation was performed on Lsd18 protein samples using a dimethylamine borane complex and acetaldehyde, right before gel filtration. For Lsd18-T189M-S195M, crystals were obtained from a 1:1 mixture of protein solution (5.5 mg·mL^−1^ in 20 mM Tris pH 8.5, 100 mM sodium chloride) and a reservoir solution (0.2 M Li_2_SO_4_, 0.1 M Tris-HCl, pH7.5, 30% *w*/*v* PEG4000) via the hanging-drop vapor diffusion method at 18 °C. The crystals were transferred into a cryoprotectant containing 20% glycerol, and flash-frozen in liquid nitrogen until X-ray data collection on beamline BL17B at the Shanghai Synchrotron Radiation Facility (Shanghai, China). For Lsd18-S195M, crystals were obtained from a 1:1 mixture of protein solution (5.5 mg·mL^−1^ in 20 mM Tris pH8.5, 100 mM sodium chloride) and a reservoir solution (0.2 M Ammonium acetate, 0.1 M BIS-TRIS pH 6.5, 25% *w*/*v* PEG3350) via the hanging-drop vapor diffusion method at 18 °C. The crystals were transferred into a cryoprotectant containing 20% glycerol, and flash-frozen in liquid nitrogen until X-ray data collection on beamline BL18U1 at the Shanghai Synchrotron Radiation Facility (Shanghai, China).

Data were processed and scaled by HKL2000 [20] and aimless [21] and the structures were solved by molecular replacement with the program MOLREP [22] using Lsd18-apo (PDB: 8UP4) as the template. Refinement was performed with Phenix.refine [23] and Refmac [24] followed by manual examination and rebuilding of the refined coordinates in the program Coot [25]. Data collection and refinement statistics are summarized in Appendix A.

## Figures and Tables

**Figure 1 ijms-24-16795-f001:**
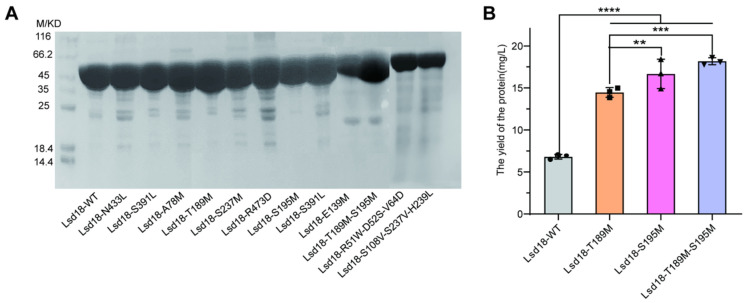
(**A**) SDS-PAGE for Lsd18-WT and mutants. (**B**) The yields for Lsd18-WT, Lsd18-T189M, Lsd18-S195M and Lsd18-T189M-S195M. ****: *p* < 0.0001, ***: *p* < 0.001, **: *p* < 0.01.

**Figure 2 ijms-24-16795-f002:**
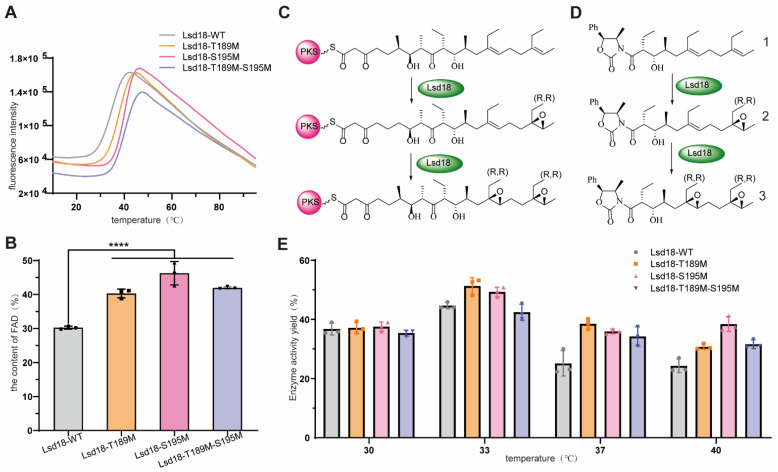
(**A**) Thermal Shift assay results for Lsd18-WT, Lsd18-T189M, Lsd18-S195M and Lsd18-T189M-S195M. (**B**) The FAD content of Lsd18-WT, Lsd18-T189M, Lsd18-S195M and Lsd18-T189M-S195M. ****: *p* < 0.0001. (**C**) The function of Lsd18 in the biosynthesis of Lasalocid A. (**D**) The substrate analog for Lsd18. 1 is the synthesized substrate analog for Lsd18, 2 is the mono-epoxide product from substrate 1 and 3 is the bis-epoxide product from substrate 1. (**E**) The catalytic activity of Lsd18-WT, Lsd18-T189M, Lsd18-S195M and Lsd18-T189M-S195M at different temperatures.

**Figure 3 ijms-24-16795-f003:**
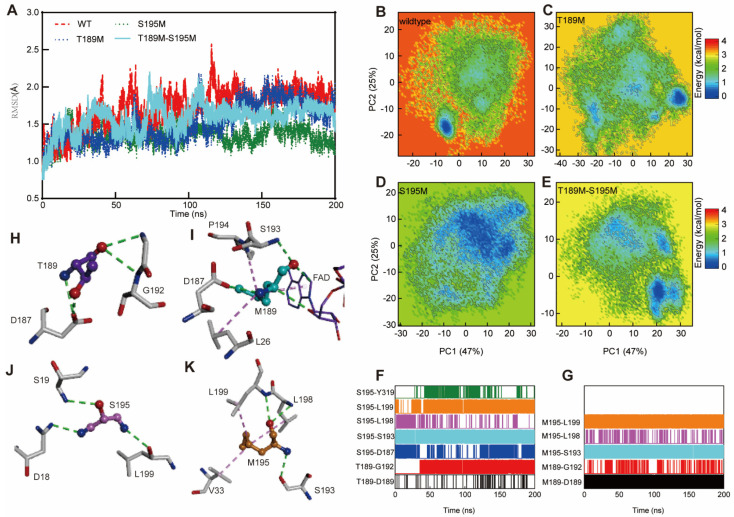
(**A**) The root–mean–square deviation (RMSD) values of Lsd18 relative to the initial structure during the 200 ns unbiased MD simulations with wild-type (in red), T189M (in blue), S195M (in green), T189M-S195M (in cyan). (**B**–**E**) Conformational landscapes of the Lsd18 protein in the (**B**) wild-type simulation, (**C**) T189M simulation, (**D**) S195M simulation, and (**E**) T189M-S195M simulation generated using PC1 and PC2 from the wild-type simulation. (**F**,**G**) The hydrogen bond profiles for T/M189 and S/M195 occurrence probabilities. (**H**–**K**) Details of the intramolecular interactions of T/M189 and S/M195 with surrounding residues in representative structures are highlighted. Key residues and T/M189 or S/M195 are represented by stick models, and ball and stick models, respectively. The O and N atoms are colored in red and blue, respectively. The important hydrogen bond and hydrophobic interactions are labeled by the dashed green and pink lines.

**Figure 4 ijms-24-16795-f004:**
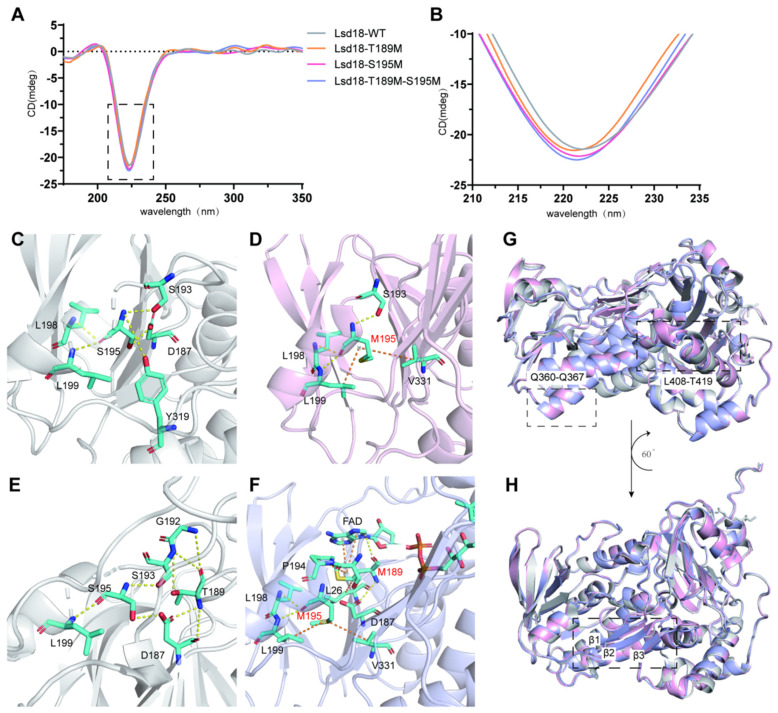
Structural analysis of Lsd18-WT and mutants. (**A**) CD spectrum of Lsd18-WT, Lsd18-T189M, Lsd18-S195M and Lsd18-T189M-S195M. (**B**) Enlarged figure of the negative peak in the CD spectrum. (**C**) The interactions between S195 and the surrounding residues in the crystal structure of Lsd18-WT. (**D**) The interactions between M195 and the surrounding residues in the crystal structure of Lsd18-S195M. (**E**) The interactions between T189 and the surrounding residues in the crystal structure of Lsd18-WT. (**F**) The interactions between M189 and the surrounding residues in the crystal structure of Lsd18-T189M-S195M. (**G**,**H**) Superposition of the structures of Lsd18-WT, Lsd18-S195M and Lsd18-T189M-S195M. The structure of Lsd18-WT is shown as a grey cartoon, the structure of Lsd18-S195M is shown as a pink cartoon and the structure of Lsd18-T189M-S195M is shown as a blue cartoon. The interaction residues are shown as cyan sticks. Hydrogen bonds are shown as yellow dashes while hydrophobic interactions are shown as red dashes.

**Table 1 ijms-24-16795-t001:** FoldX calculated mutants.

Mutant	ΔΔG (kcal/mol)
S195M	−4.59
N433L	−3.49
A78M	−3.49
T189M	−3.37
S237M	−3.19
R473D	−3.00
E139M	−2.91
S391L	−2.73
G101P	−2.57

**Table 2 ijms-24-16795-t002:** Rosetta calculated mutants.

Mutant	ΔΔG (kcal/mol)
P326A	−9.38
P326S	−7.06
R21G	−8.85

**Table 3 ijms-24-16795-t003:** Lsd18 mutagenesis primer design.

Primer	Sequence (5′→3′)
Lsd18-F-NdeI	GGAATTCCATATGACGAACACGCGCTC
Lsd18-R-EcoRI	CCGGAATTCTTAAGCGGTAACACCAG
S195M-F	CGCGGTAGCCCGATGAAACGTCTG
S195M-R	CAGACGTTTCATCGGGCTACCGCG
N433L-F	CAGGTTGTCCTTCGTGCAGCTGTG
N433L-R	CACAGCTGCACGAAGGACAACCTG
A78M-F	GTCTGGCGGTATGCGTATCGTTGAAG
A78M-R	CTTCAACGATACGCATACCGCCAGAC
T189M-F	GATCTGGTGGTTGACACCATGGGTCG
T189M-R	CGACCCATGGTGTCAACCACCAGATC
S237M-F	CTTCCCGCTGGTTATGGTCCATGC
S237M-R	GCATGGACCATAACCAGCGGGAAG
R473D-F	GAACCGCCGCTGGACCCGGATGAAG
R473D-R	CTTCATCCGGGTCCAGCGGCGGTTC
E139M-F	CGTTCGTGAAATGACGCTGCGTGAAG
E139M-R	CTTCACGCAGCGTCATTTCACGAACG
S391L-F	GTGCTGGCAACCCTTCATGATATC
S391L-R	GATATCATGAAGGGTTGCCAGCAC
G101P-F	GCTCATCGTATTCCCATCCCGGAC
G101P-R	GTCCGGGATGGGAATACGATGAGC
R21G-F	GGAATTCCATATGACGAACACGGGCTCGGCGGT
R21G-R	CCGGAATTCTTAAGCGGTAACACCAG
P326A-F	CTGGCAACCTGGGCGGAAGGCCTG
P326A-R	CAGGCCTTCCGCCCAGGTTGCCAG
P326S-F	CTGGCAACCTGGTCGGAAGGCCTG
P326S-R	CAGGCCTTCCGACCAGGTTGCCAG
R51W-D52S-F	CGTTATTGATTGGAGCGCATTTCCGG
R51W-D52S-R	CCGGAAATGCGCTCCAATCAATAACG
V64D-F	GCGCAAAGGTGACCCGCAAGCTCG
V64D-R	CGAGCTTGCGGGTCACCTTTGCGC
S237V-H239L-F	CCGCTGGTTGTGGTCCTTGCTGATCAC
S237V-H239L-R	GTGATCAGCAAGGACCACAACCAGCGG
S108V-F	CGGTCAGGTGGTTTATACCGC
S108V-R	GCGGTATAAACCACCTGACCG

**Table 4 ijms-24-16795-t004:** Thermal Shift Assay reaction system.

Reagent	Amount
50× Sypro-Orange	6 μL
4× Tris-HCl buffer	15 μL
Lsd18 WT/mutants	12 μg
ddH_2_O	Add to 60 μL

**Table 5 ijms-24-16795-t005:** Enzyme activity reaction system.

Reagent	Concentration	Amount
Buffer	4×	50 μL
substrate	4 mM	10 μL
NADH	100 mM	4 μL
NADPH	100 mM	4 μL
FAD	10 mM	1.6 μL
Enzyme	20 mg/mL	8.5 μL
Fre	15 μM	16.4 μL
Methanol	30%	55 μL
MQ		50.5 μL

**Table 6 ijms-24-16795-t006:** LC-MS program.

Time	A, B Ratio	Speed
0 min	20%A, 80%B	0.2 mL/min
0.5 min	20%A, 80%B	0.2 mL/min
6 min	0%A, 100%B	0.2 mL/min
6.5 min	0%A, 100%B	0.2 mL/min
8 min	20%A, 80%B	0.2 mL/min

## Data Availability

The data presented in this study are available in the main text. Further information are available on request from the corresponding authors.

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
