# Peer review of "Simultaneous Improvement in the Thermostability and Catalytic Activity of Epoxidase Lsd18 for the Synthesis of Lasalocid A"

_ijms, 2023, doi:10.3390/ijms242316795_

Round 1
Reviewer 1 Report
Comments and Suggestions for Authors
In the manuscript entitled “Simultaneous improvement of the thermostability and catalytic activity of epoxidase Lsd18 for the synthesis of Lasalocid A” by Ning Liu et al., the authors report the in-silico guided protein engineering of the epoxidase Lsd18 to improve its thermostability and catalytic activity. The manuscript is well written, and the authors conclusions are generally supported by the obtained results. However, some conclusions are not. Therefore, I endorse publications of the manuscript in IJMS after minor revision.
Major points:
Page 5, Rows 147-151, authors state: “Remarkably, at a reaction temperature of 40℃, Lsd18-S195M exhibited the highest activity among the wild-type Lsd18 and the other two mutants. Collectively, these results demonstrate that our engineering strategy not only enhanced Lsd18’s stability but also optimized its catalytic activity at elevated reaction temperatures.”. This cannot be said, because the results do not support this hypothesis.
Instead, my suggestion is to state that: “At a reaction temperature of 40℃, Lsd18-S195M exhibited an activity comparable to that of the wild-type enzyme at 40℃. Collectively, these results demonstrate that our engineering strategy has enhanced Lsd18’s thermal stability.”
Page 6, Rows 192-195, authors state: “Before crystallography study, we also performed CD analyses to compare the secondary structures of Lsd18, Lsd18-T189M, Lsd18-S195M and Lsd18-T189M-S195M. As shown in Fig. 4A-B, there was a minor difference in the negative peak, which indicated that the mutants were more structured than the wild-type protein.”. This cannot be said, because the results do not support the author’s conclusions. The slight difference shown in Fig. 4A-B, cannot support the statement that the mutant are more structured than the wild-type protein. Indeed, such small difference can be due, for instance, to slight difference in the concentrations of the protein samples tested.
What can be stated, instead, is: “Before crystallography study, we also performed CD analyses to compare the secondary structures of Lsd18, Lsd18-T189M, Lsd18-S195M and Lsd18-T189M-S195M. As shown in Fig. 4A-B, no apparent differences in the negative peak are observed, which indicated that the mutants were as well-structured as the wild-type protein.”.
Page 6, Rows 214-216, authors state: “More importantly, M189 forms a hydrogen bond with the cofactor FAD, which could stabilize the binding of this cofactor, thus promote the enzymatic activity of this mutant.”. This cannot be stated, because the results do not support the author’s conclusions.
Instead, my suggestion is to state that: “More importantly, M189 forms a hydrogen bond with the cofactor FAD that could stabilize the binding of this cofactor. Therefore, it is speculated that this could improve the enzymatic activity of this mutant.”.
Page 6, Rows 214-216, authors state: “Our crystal structures of the mutants showed that stabilization could lead to more rigid enzymes, which might not be beneficial for the activity of the enzyme. However, at higher temperature, the mutants could overcome this rigidity and be more active in catalysis. Although there is a dilemma between the stability and activity, our example shows that to improve the stability of the enzyme could also improve the catalytic activity.”. This cannot be said, because the results do not support the author’s conclusions.
Instead, my suggestion is to state that: “Our crystal structures of the mutants showed that stabilization could lead to more rigid enzymes, which might not be beneficial for the activity of the enzyme. However, at higher temperature, the mutants could overcome this rigidity and be as catalytically active as the wild-type enzyme at lower temperature. Although there is a dilemma between the stability and activity, our example shows that to improve the stability of the enzyme could help mutated enzymes to retains their catalytic activity at higher temperature.”.
Minor points:
Check the caption of Figure 4B-3.
Comments on the Quality of English Language
Minor editing of English language required.
Reviewer 2 Report
Comments and Suggestions for Authors
The authors have very nice results of protein engineering by the use of FoldX and Rosetta software. The results are supported partly by their following structural studies.
1. I should like to point out the authors are undoubtfully using the FoldX and Rosetta software. However, the authors should discuss the effectiveness and reliabilities compared from other methods or software, citing appropriate review article original papers.
2. Furthermore, there is a drawback that important kinetic parameters are not measured with the variants. Therefore, at least a discussion has to be made if the variants can be utilized for the purpose of the authors. The authors state in the introduction that "improving its thermostability is the first step to promote its application in organic and industrial synthesis of epoxides."
